# The STAT Signaling Pathway in HIV-1 Infection: Roles and Dysregulation

**DOI:** 10.3390/ijms26189123

**Published:** 2025-09-18

**Authors:** Manlio Tolomeo, Antonio Cascio

**Affiliations:** 1Department of Health Promotion Sciences, Maternal and Infant Care, Internal Medicine and Medical Specialties, University of Palermo, 90127 Palermo, Italy; antonio.cascio03@unipa.it; 2Department of Infectious Diseases, Azienda Ospedaliera Universitaria Policlinico (A.O.U.P.) Palermo, 90127 Palermo, Italy

**Keywords:** HIV-1, STAT signaling pathway, immune dysregulation, T helper subsets, viral latency, AIDS, HIV-1 therapeutic approaches

## Abstract

The STAT (Signal Transducer and Activator of Transcription) signaling pathway plays a central role in immune regulation by mediating cytokine responses and orchestrating both innate and adaptive immunity. Although CD4+ T cell depletion is the main driver of HIV-1–induced immunodeficiency, the virus also exerts a significant and often underestimated impact by disrupting the function of STAT family members, thereby exacerbating immune imbalance and accelerating disease progression. Specifically, HIV-1 suppresses STAT1 activation, impairing the induction of antiviral genes; inhibits IL-23–driven STAT3 activation in CD4+ Th17 cells with a reduction in IL-17; alters STAT3-dependent functions in antigen-presenting cells; and imposes profound—and at times opposing—dysregulations of STAT5, including the induction of a truncated isoform that contributes to latency. Notably, pharmacological inhibition of the JAK/STAT axis, particularly with JAK2 inhibitors, has been shown to reduce integrated proviral DNA and viral replication in vitro and in early clinical studies. This review provides an updated overview of the roles of individual STAT proteins in HIV-1 infection and pathogenesis, emphasizing the intricate interplay between viral factors and host signaling, highlighting the potential therapeutic implications, and suggesting that immunological assessment in HIV-1 patients should extend beyond CD4+ T cell counts and the CD4/CD8 ratio to include functional analysis of STAT signaling for deeper insights into immune dysfunction and chronic inflammation.

## 1. Introduction

Signal Transducers and Activators of Transcription (STATs) are a family of cytoplasmic transcription factors that play a pivotal role in transmitting signals from cell surface receptors to the nucleus in response to cytokines and growth factors. Upon phosphorylation by Janus kinases (JAKs), STAT proteins dimerize, translocate to the nucleus, and regulate genes involved in immunity, inflammation, cell proliferation, and apoptosis [1]. The JAK/STAT pathway is evolutionarily conserved and functionally versatile, with individual STAT family members (STAT1–STAT6) exerting context-dependent roles in antiviral responses, tumor progression, and immune cell polarization [1].

Human Immunodeficiency Virus type 1 (HIV-1), the etiological agent of AIDS, primarily targets CD4+ T cells, macrophages, and dendritic cells, leading to progressive immune dysfunction and increased susceptibility to opportunistic infections and malignancies [2]. Despite the success of antiretroviral therapy (ART) in suppressing viral replication, HIV-1 persists in long-lived cellular reservoirs, posing a formidable barrier to viral eradication [3]. The virus manipulates host immune signaling pathways to facilitate replication, evade innate immunity, and induce chronic immune activation [2,3,4].

Among the host signaling cascades affected by HIV-1, the JAK/STAT pathway has emerged as a critical axis. HIV-1 inhibits interferon (IFN)–mediated antiviral responses by degrading components of the JAK/STAT cascade [5] or modulating STAT activity, thereby promoting immune evasion and enhancing viral replication [6,7]. Chronic activation of certain STAT family members has also been implicated, directly or indirectly, in HIV-associated immune exhaustion, mucosal barrier dysfunction, and lymphoid tissue fibrosis [8,9,10,11].

Despite accumulating evidence, the precise roles of individual STAT proteins and their upstream JAK kinases in HIV-1 infection remain incompletely understood. Many experimental findings are challenging to interpret and, at times, yield apparently contradictory results. Nevertheless, HIV-1 plays a central and often underappreciated role in driving substantial immune suppression through the dysregulation of STAT family members. This dysregulation occurs alongside CD4+ T cell depletion and contributes to exacerbated chronic inflammation, a key factor in immunoaging among people living with HIV-1. Importantly, these STAT-related alterations frequently persist in patients on ART, even after CD4+ T cell recovery.

Given these gaps in understanding, this review provides an updated overview of the JAK/STAT signaling pathway in HIV-1 infection, with a particular focus on individual STAT family members. Drawing on recent advances and integrating insights from our previous work on the STAT system [12,13,14,15,16,17,18], we explore the complex interplay between HIV-1 and host STAT signaling, highlighting both the pathogenic consequences of STAT dysregulation and its potential as a therapeutic target.

## 2. The JAK/STAT Pathway

The Janus kinase/signal transducer and activator of transcription (JAK/STAT) pathway is a central signaling cascade that mediates cellular responses to a wide variety of extracellular cues, including cytokines, interferons, hormones, and growth factors [19,20,21]. This pathway is one of the most evolutionarily conserved and rapid mechanisms for signal transduction from the cell membrane to the nucleus. It enables cells to respond dynamically to environmental and immunological stimuli. The core components of this pathway include four JAK family kinases (JAK1, JAK2, JAK3, and TYK2) and seven STAT transcription factors (STAT1, STAT2, STAT3, STAT4, STAT6, with two isoforms of STAT5: STAT5A and STAT5B) [12,13,14,15,22,23,24].

Signal initiation occurs when a ligand binds to its cognate receptor, inducing receptor dimerization and activation of receptor-associated JAKs. Upon activation, JAKs phosphorylate specific tyrosine residues within the cytoplasmic domains of the receptors, creating docking sites for the SH2 domains of STAT proteins. Upon recruitment, STATs are themselves phosphorylated by JAKs, leading to dimerization via reciprocal SH2-phosphotyrosine interactions. The resulting dimers translocate to the nucleus, where they bind specific DNA elements to regulate gene expression programs involved in cell survival, proliferation, differentiation, and immune function [1,12,13,14,15,22,23,24,25].

Each STAT protein plays both unique and overlapping roles in immune homeostasis and host defense. STAT1 is activated by type I, II, and III IFNs and is pivotal in antiviral immunity, inducing hundreds of interferon-stimulated genes (ISGs) [12,26,27]. STAT2 is primarily activated by type I and III IFNs, functioning in concert with STAT1 and interferon regulatory factor 9 (IRF9) in the formation of the ISGF3 (Interferon Stimulated Gene Factor 3), which transactivates ISGs with ISRE (IFN-stimulated response elements) motifs [12,27]. STAT3 responds to IL-6 family cytokines and regulates inflammation, cell survival, and immune tolerance; dysregulated STAT3 signaling is implicated in both cancer and chronic infection [13,28,29]. STAT4 and STAT6 are crucial for T helper cell differentiation: STAT4 promotes Th1 polarization in response to IL-12, while STAT6 drives Th2 responses via IL-4 and IL-13 [14,30,31]. STAT5A and STAT5B, activated by cytokines such as IL-2, IL-7, and IL-15, regulate lymphocyte development, homeostasis, and cell proliferation [15,32].

The STAT pathway is tightly regulated by a range of negative feedback mechanisms, including suppressors of cytokine signaling (SOCS), protein inhibitors of activated STATs (PIAS), and ubiquitin-specific proteases such as USP18. These mechanisms ensure a balance between immune activation and resolution, thus maintaining immune equilibrium and preventing pathological inflammation or autoimmunity [23,26,33].

Beyond its immunological relevance, aberrations in JAK/STAT signaling have been linked to numerous diseases, including autoimmune syndromes, hematological malignancies, solid tumors, and chronic viral infections [12,13,14,15].

## 3. HIV-1 and Host STAT Pathways

Since its emergence in the early 1980s, HIV/AIDS has posed one of the most complex and devastating global health challenges. HIV-1 primarily targets CD4+ T lymphocytes, leading to a gradual collapse of cell-mediated immunity and increased susceptibility to opportunistic infections and neoplasms [34,35]. Despite the success of ART in controlling viremia and restoring immune function, HIV-1 remains incurable due to the persistence of latent reservoirs, chronic immune activation, and tissue-level dysfunction [3,36,37].

HIV-1 is an RNA retrovirus capable of synthesizing double-stranded linear viral DNA (vDNA) from its RNA genome via reverse transcription, catalyzed by the viral enzyme reverse transcriptase (RT) [38,39,40]. Integration of vDNA into the host genome is a critical and irreversible step, committing the cell to permanently harbor the viral genome as a provirus. HIV-1 proviral integration, or viral regulatory and accessory proteins, can interact with the host genome, causing DNA damage or gene dysregulation [41]. The acute phase of infection is marked by rapid viral replication and massive CD4+ T cell depletion in gut-associated lymphoid tissue. As infection progresses, chronic immune activation and inflammation persist, even under suppressive ART [42]. Multiple viral proteins—including Tat, Nef, Vif, and gp120—modulate apoptosis, antigen presentation, cytokine production, and signal transduction [43,44]. The virus also evolves mechanisms to subvert innate immunity, particularly type I IFN responses, to evade detection and establish persistent infection [45].

HIV-1 infection dynamically rewires host cytokine signaling, alters T cell subset distributions, and dysregulates transcriptional programs that maintain immune homeostasis. These effects highlight the central role of host pathways such as JAK/STAT, which may exert both protective and pathogenic functions during infection [2,3,4,5,6,7,8,9,10,11,35,36,37,38,39,40,41].

Recent studies reveal a multifaceted involvement of the STAT pathway in HIV-1 infection. Intact JAK/STAT signaling—particularly via STAT1, STAT3, and STAT5—is essential for initiating antiviral responses and maintaining lymphocyte function. Conversely, the virus can exploit or inhibit STAT family members to escape immune surveillance, sustain inflammation, and preserve viral reservoirs (Figure 1) [46,47,48]. STAT proteins mediate these effects through modulation of IFN signaling, inflammatory cytokine responses, and T cell differentiation. The balance between protective and pathogenic STAT functions is context-dependent, varying across isoforms and stages of infection.

Given the role of STAT signaling in HIV-1 infection, pharmacological modulation of this pathway has emerged as a potential therapeutic strategy. JAK/STAT inhibitors, such as ruxolitinib, are under investigation as adjunctive treatments in HIV-1 infection [49,50]. Additional STAT inhibitors, currently studied for other diseases, may in the future serve as promising adjunctive therapies alongside conventional ART [51,52].

## 4. The Impact of HIV-1 on STAT1 Signaling and Pathway Dysregulation

STAT1 is a central mediator of IFN signaling and plays a critical role in antiviral immunity [12]. Type I IFNs, including IFN-α and IFN-β, are released by infected cells to trigger an innate immune response that limits viral replication. This response is mediated through activation of the JAK–STAT signaling pathway, which results in phosphorylation of STAT1 at both Tyr701 and Ser727 residues by JAKs. Phosphorylated STAT1 forms homodimers and heterodimers (primarily with STAT2), which translocate to the nucleus and drive the transcription of ISGs that exert antiviral effects by encoding antiviral proteins (Table 1) [12,53,54,55]. HIV-1 infection profoundly dysregulates STAT1, with major implications for immune function and disease pathogenesis. In contrast, no specific or relevant data regarding STAT2 and HIV-1 are reported in the literature.

### 4.1. HIV-1–Mediated Modulation of STAT1: Context-Dependent Mechanisms

Divergent findings have been reported in different studies conducted by various authors regarding the effects of HIV-1 on STAT1. Federico et al. showed that HIV-1 Nef protein selectively stimulates both the α and β isoforms of STAT1. This effect was demonstrated using two complementary approaches: infection of human monocyte-derived macrophages with HIV-1 mutants lacking specific genes and exposure of monocyte-derived macrophages to exogenous recombinant Nef protein (rNef), which these cells are capable of internalizing. The Nef-driven activation of STAT1 is subsequently associated with elevated levels of STAT1 itself and of IRF-1, a downstream transcription factor whose expression is under the control of STAT1 signaling [56].

Similar results were obtained by Kohler et al., who examined the interaction between HIV-1 and CD4+ cells by analyzing phosphorylated STAT proteins in nuclear extracts from lymphoid and monocytic cell lines, as well as primary monocyte-derived macrophages. Their findings revealed a marked increase—ranging from six- to tenfold—in activated STAT1, STAT3, and STAT5 in HUT78 and U937 cells within two hours of exposure to HIV-1 virions. Notably, STAT activation was triggered by various viral envelope proteins and occurred regardless of viral coreceptor usage or replication capacity, whereas envelope-deficient particles failed to induce any STAT activation [57].

In contrast to the findings reported by Federico and Kohler, Nguyen et al. observed that HIV-1 impairs STAT1 activation in T lymphoblastic cells transfected with HIV plasmids, resulting in diminished ISG expression and compromised antiviral defense [58]. They observed that HIV-1 accessory proteins Vpu and Nef play a central role in suppressing IFN-α–mediated STAT1 phosphorylation, effectively allowing the virus to evade immune detection and clearance. This impairment facilitates more efficient viral replication, even in the presence of circulating type I interferons, and contributes to persistent infection and progression toward AIDS. Although the precise mechanisms by which HIV-1 disrupts STAT1 activation remain to be fully elucidated, these authors hypothesize that Vpu and Nef may interfere with the kinase activity of Tyk2 or Jak1—critical components of the JAK–STAT cascade—without necessarily affecting their expression levels (Figure 2). In addition, Gargan et al. [59] demonstrated that peripheral blood mononuclear cells (PBMCs) from patients with HIV have lower STAT1 and STAT3 protein levels, impaired IFN-α-mediated phosphorylation, and reduced ISG15 induction. They found Vif interacts with STAT1 (but not STAT2), and its expression promotes ubiquitination and MG132-sensitive, proteosomal degradation of this transcription factor. (Figure 2). This degradation process requires Vif’s Elongin-Cullin-SOCS-box (ECS) binding motif, which facilitates the assembly of an active E3 ubiquitin ligase complex. Notably, the scaffold proteins Cul5 and Rbx2 were also shown to be essential for Vif-induced degradation of STAT1 (Figure 2) [59].

The apparently contradictory findings regarding HIV-1’s effects on STAT1 may be reconciled by considering the context, timing, and cellular environment of these effects.

HIV-1 exerts cell-type- and context-dependent modulation of STAT1. Early after exposure, HIV-1 envelope proteins and accessory proteins can induce transient STAT1 activation in monocytes/macrophages, leading to phosphorylation at Tyr701 and Ser727 and upregulation of inflammatory genes. In contrast, during or after productive infection, HIV-1 proteins (notably Vpu, Nef, and Vif) actively inhibit canonical type I IFN signaling by blocking IFN-α-induced STAT1 phosphorylation and promoting proteasomal degradation of STAT1. This results in type I IFN refractoriness, a state where exogenous IFN-α fails to induce STAT1 phosphorylation and ISG expression, thereby blunting the antiviral response.

Thus, HIV-1 may initially activate STAT1 through non-IFN pathways to promote inflammation and transient antiviral activity but may later suppress IFN-driven STAT1 activation to evade the host antiviral response. Consistent with this hypothesis, Federico et al. reported that while Nef initially triggers STAT1 activation, this activation progressively declines over time in both Nef-treated and HIV-1–infected cells, despite Nef levels remaining stable in infected cells. Although direct in vivo confirmation of this temporal sequence is limited, and few studies have systematically investigated this dynamic, the proposed model offers a plausible framework to reconcile the divergent findings reported across different experimental contexts [56,57,58,59].

Despite sustained expression of antiviral type I interferons (IFNs), HIV-1 infection persists in humans, and even exogenous IFN-α has only a modest effect on viral load, suggesting that people living with HIV-1 may develop refractoriness to type I IFN. Sugawara et al. provided evidence for this phenomenon by showing that CD4+ and CD8+ T cells from both untreated and virologically suppressed HIV-1-infected patients display impaired STAT1 phosphorylation and diminished induction of ISGs following IFN stimulation, compared to healthy controls. These refractory responses were associated with markedly elevated levels of USP18 in the same T cell subsets, a finding further validated in an independent cohort of HIV–HCV co-infected individuals receiving pegylated IFN-α2b [60]. USP18 (ubiquitin-specific peptidase 18) is a well-known negative regulator of type I interferon signaling. Following interferon stimulation, USP18 is induced and functions as a brake on the pathway. It exerts this effect by binding to IFNAR2, one of the two subunits of the type I interferon receptor (together with IFNAR1) that normally initiate JAK/STAT signaling. By occupying IFNAR2, USP18 prevents the recruitment of JAK1, thereby shutting down the signaling cascade and avoiding uncontrolled pathway activation. Supporting this mechanism, in vitro experiments showed that conditioned media from HIV-1–infected PBMCs can induce refractoriness in uninfected CD4+ T cells, an effect partially reversed by USP18 knockdown. Collectively, these findings suggest a model in which persistent type I IFN production during HIV-1 infection upregulates USP18, creating a negative-feedback loop that attenuates subsequent IFN responses.

Beyond viral interference, host factors can also downregulate STAT1 activation. Rivera et al. reported that cystatin B, a cysteine protease inhibitor, impairs STAT1 signaling and IFN-β-mediated antiviral responses, thereby promoting HIV-1 persistence in macrophage reservoirs [61]. In monocyte-derived macrophages, cystatin B enhances HIV-1 replication and interacts with STAT1, inhibiting its nuclear translocation in Vero cells. It also reduces tyrosine-phosphorylated STAT1 levels and limits its transport to the nucleus. Concurrently, HIV-1 infection retains unphosphorylated STAT1 within the nucleus, preventing its export to the cytoplasm for subsequent phosphorylation [61]. This combined viral and cellular inhibition of STAT1 can contribute to viral persistence in key reservoirs.

### 4.2. Atypical Activation of ISG and STAT1 by HIV-1 Tat Protein

HIV-1 Tat can upregulate STAT1 expression independently of IFNs, representing an atypical mechanism that modulates immune signaling, primes cells for immune responses without inducing a full antiviral state, and subtly alters antigen presentation and T cell activation. Kukkonen et al. [62] demonstrated that HIV-1 Tat directly regulates the expression of ISGs, including STAT1, in antigen-presenting cells (APCs) by interacting with specific gene promoters and transcription factors. This process bypasses the canonical IFN-induced JAK-STAT pathway, in which viral recognition triggers IFN production followed by ISG activation [62]. Tat activation of ISG is mediated by Tat association with the promoters of *MAP2K6* and *MAP2K3*—which activate p38MAPK (p38 mitogen-activated protein kinase)—and of IRF7.

IRF7 can activate ISGs independently of the IFN-JAK-STAT pathway. IRF7 is a master transcription factor in the innate immune response, particularly in the induction of type I IFNs and a subset of ISGs. Upon activation by pathogen recognition receptor (PRR) signaling—such as through TLR7/9 in plasmacytoid dendritic cells or RIG-I/MDA5 in other cell types—IRF7 is phosphorylated, dimerizes, translocates to the nucleus, and binds to ISREs in the promoters of ISGs, directly driving their transcription [62].

Tat’s ability to activate ISGs independently of IFNs highlights a strategy by which HIV-1 hijacks components of the host antiviral defense while attenuating their full efficacy. The Tat protein is composed of two exons: exon 1 is essential for viral transactivation and interaction with host factors, whereas exon 2 is dispensable for transactivation. Interestingly, truncated Tat variants lacking exon 2 elicit stronger ISG expression than the full-length protein, suggesting that exon 2 functions as a regulatory domain that restrains Tat-induced immune activation. This implies that HIV-1 fine-tunes ISG expression through Tat to avoid initiating a robust antiviral state typically triggered by IFNs. Such modulation may enable the virus to induce a partial immune response, strong enough to mimic cellular activation, yet insufficient to promote effective viral clearance. In this context, the immunomodulatory role of exon 2 may be critical in shaping a host environment conducive to viral persistence or latency, particularly within APCs (Figure 3) [62].

A comparative analysis of the studies by Nguyen et al. [58] and Kukkonen et al. [62] highlights distinct yet complementary mechanisms through which HIV-1 manipulates STAT1 signaling to undermine host antiviral defenses. Collectively, the findings suggest that HIV-1 adopts a dual strategy: on the one hand, it suppresses STAT1 phosphorylation via viral proteins such as Vpu and Nef, thereby dampening type I IFN–mediated antiviral responses; on the other, it exploits Tat to modulate—rather than fully suppress—STAT1-dependent gene expression. This selective modulation contributes to persistent immune activation without triggering an effective antiviral state. Such layered interference with STAT1 signaling may allow the virus to evade early immune responses while fostering a chronically activated yet permissive immune environment, favorable to the establishment and persistence of viral reservoirs.

Elucidating these diverse mechanisms opens new avenues for therapeutic intervention. Strategies could include targeting Vpu and Nef to restore JAK/STAT pathway responsiveness, inhibiting cystatin B to reinstate effective STAT1 activation, or blocking Tat-mediated modulation to re-establish a robust interferon-induced antiviral state. Such approaches hold the potential to enhance the efficacy of type I IFNs, counteract HIV-mediated immune evasion, and ultimately limit viral replication and reservoir persistence.

## 5. HIV-1 and STAT3: Molecular Interactions and Pathophysiological Implications

STAT3 is a critical transcription factor involved in regulating immune responses, cell proliferation, and survival [63,64]. HIV-1 interacts with the host cell machinery to facilitate its replication and persistence, and accumulating evidence suggests that HIV-1 modulates STAT3 activity at multiple levels. STAT3 plays a central role in cytokine signaling pathways (particularly IL-6 and IL-10), and type I IFNs [13,63], which are dysregulated in HIV-1 infection. HIV-1 modulates STAT3 signaling through a variety of mechanisms, including cytokine induction, viral protein interaction, and modulation of regulatory pathways [9,65,66]. This interaction contributes to viral persistence, immune dysregulation, and disease progression.

### 5.1. HIV-1 gp120-Mediated Activation of the STAT3/IL-6 Axis in Dendritic Cells

Dendritic cells are central to both the initiation of antiviral immunity and the pathogenesis of HIV-1 infection. Their ability to sense, capture, and present antigens makes them indispensable for immune surveillance. STAT3 activation in dendritic cells suppresses their immunostimulatory function and promotes immune tolerance [9]. This has implications in cancer, autoimmunity, and chronic viral infections like HIV-1, where it contributes to immune dysfunction and pathogen persistence [9,67].

HIV-1 gp120 plays a pivotal role in modulating dendritic cell function through the activation of the STAT3/interleukin-6 (IL-6) signaling axis in primary human monocyte-derived dendritic cells [9]. Upon exposure to gp120, dendritic cells rapidly produce IL-6 in a CCR5 (C-C chemokine receptor type 5)-dependent manner. This induction requires the activation of both the MAPK and nuclear factor kappa B (NF-κB) pathways, which mediate IL-6 transcription and release. The secreted IL-6 then acts in an autocrine fashion to activate STAT3 through tyrosine phosphorylation, establishing a classical feedback loop (Figure 4A) [9].

Remarkably, gp120 also induces an early, IL-6-independent phosphorylation of STAT3, which precedes IL-6 production and serves to amplify it [9]. This biphasic STAT3 activation results in a feed-forward mechanism that sustains STAT3 signaling over time. Concurrently, gp120 upregulates the expression of protein inhibitor of activated STAT3 (PIAS3), a known negative regulator of STAT3, indicating the presence of a compensatory feedback mechanism aimed at limiting excessive activation. However, the overall effect remains a prolonged activation of STAT3. Notably, this signaling cascade is not triggered by CCL4, the natural CCR5 ligand, underscoring the specificity of the response to the viral envelope protein gp120 [9].

Persistent STAT3 activation in dendritic cells has profound immunological consequences, as it interferes with their maturation, antigen-presenting capacity, and ability to activate naive T cells [9,67]. Through the targeted activation of STAT3 and modulation of IL-6 signaling, HIV-1 gp120 compromises the immunostimulatory function of dendritic cells, contributing to early immune dysfunction and viral immune evasion. Importantly, this process occurs without requiring productive infection of dendritic cells, highlighting a strategic mechanism by which HIV-1 subverts host innate immunity and facilitates viral persistence [9].

### 5.2. Role of STAT3 in Th17 and Treg Differentiation in HIV-1 Infection

Th17 and Treg cells are critical targets of HIV-1, and their respective depletion (Th17) and dysfunction (Tregs) significantly contribute to disease progression toward AIDS [68,69,70,71,72,73,74]. The interplay between these two CD4+ T cell subsets is both dynamic and complex.

Th17 cells are essential for mucosal immunity and play a protective role against opportunistic infections [35]. Their marked reduction during HIV-1 infection is strongly linked to the development of such infections, which remain a leading cause of mortality in individuals with AIDS [72,73,74]. Preserving the number and function of Th17 cells is, therefore, a key therapeutic objective in HIV-1 management.

Treg cells, by contrast, are immunosuppressive CD4+ T cells that follow a different trajectory during infection. In people with well-controlled HIV-1 under antiretroviral therapy, Tregs have been shown to contain a higher proportion of inducible, genetically intact proviruses compared to other CD4+ subsets [75]. Although their absolute counts may remain stable or decline slowly, their suppressive capacity often increases substantially during infection, aggravating immune dysfunction.

Notably, Th17 and Treg cells exhibit functional plasticity. Under specific cytokine conditions, they can transdifferentiate into one another. In murine models, for example, IL-6 can convert Tregs into Th17 cells in the absence of TGF-β [76,77]. Similar interconversion has been observed in humans, depending on the immunological context and cytokine environment [78,79].

STAT3 is a central regulator of both Th17 and Treg differentiation, and its dysregulation in HIV-1 infection plays a key role in the imbalance of these subsets [35]. To become fully functional Th17 cells, immature T cells need to receive signals from IL-6, IL-21, and IL-23. This process transforms them into cells that produce inflammatory molecules like IL-17, GM-CSF, and IFN-γ [80]. Cytokines that induce the development of Th17 cells all activate STAT3. STAT3 binds directly to the promoter and enhancer regions of the *RORC* (*retinoic acid-related orphan receptor C)* gene, promoting the expression of the master transcription factor RORγt [81]. STAT3 also induces epigenetic remodeling of chromatin accessibility at the *RORC* locus and other Th17-related genes. It recruits histone-modifying enzymes that add activating marks (e.g., H3K4me3) and remove repressive ones (e.g., H3K27me3) [81]. RORγt binds specific DNA sequences in the promoter and enhancer regions of Th17 signature genes, most notably: *Il17a* and *Il17f* (genes encoding IL-17A and IL-17F), *Il23r* (IL-23 receptor), *Csf2* (encoding GM-CSF), and *RORC* itself (positive feedback loop). By binding to these loci, RORγt recruits co-activators and promotes transcription of these genes, especially *Il-17A/F*, which are hallmark cytokines of Th17 cells [81].

In the absence of strong STAT3 signaling, Foxp3 (which promotes Treg development) is upregulated and can directly inhibit RORγt function. STAT3 suppresses Foxp3 and thus indirectly protects RORγt from inhibition, reinforcing the Th17 lineage over Treg fate. This makes the *RORC* gene more transcriptionally active and its expression more stable. Treg development is primarily induced by TGF-β and FOXP3, and STAT3 activation can counteract Treg induction under certain cytokine conditions [80,81,82,83].

In the context of HIV-1 infection, Th17 cells are preferentially lost, while Treg frequencies are often elevated. This imbalance contributes to mucosal barrier disruption and chronic immune activation [35,84,85,86]. Unlike in dendritic cells—where HIV-1 gp120 promotes STAT3 activation—HIV-1 impairs IL-23-induced STAT3 phosphorylation in Th17 cells (Figure 4B) [87]. Fernandes et al. investigated how HIV infection affects IL-23-mediated responses in Th17 cells using an in vitro model. Specifically, they assessed IL-17 production, phosphorylation of STAT3 (pSTAT3), and expression of the *RORC* gene. They also analyzed blood-derived Th17 cells from both untreated and ART-treated HIV-1-positive individuals to evaluate IL-23-induced STAT3 phosphorylation and IL-23 receptor expression. Their results demonstrated that HIV infection markedly reduced IL-17 production and IL-23-stimulated pSTAT3 levels in vitro, while *RORC* transcription remained unchanged. Moreover, Th17 cells from both untreated and ART-treated individuals showed a complete absence of IL-23-induced pSTAT3, despite preserved expression of IL-23 receptors. This indicates that HIV disrupts IL-23 signaling downstream of receptor engagement, an effect not reversed by ART. These findings provide a possible explanation for the persistent immune activation and incomplete restoration of Th17 cell function despite effective viral suppression with ART [87].

Altogether, the disruption of STAT3 signaling by HIV-1—through interference with cytokine responses and altered dendritic cell activity—represents a key mechanism underlying the immune dysregulation seen in infected individuals.

### 5.3. Insertional Mutagenesis and Oncogenesis

HIV-1 DNA integration at the STAT3 locus in primary human CD4+ T cells directly drives T cell persistence and models the molecular pathogenesis of HIV-associated T cell lymphoma. Using an in vitro infection system, Rist et al. identified that HIV-1 proviral integration within intron 1 of the STAT3 gene leads to upregulation of STAT3 expression and activation of its downstream transcriptional program. This integration event results in a persistent population of HIV-infected T cells with a STAT3-driven gene signature characterized by enhanced anti-apoptotic phenotypes, closely resembling those observed in HIV-associated T cell lymphomas [88]. Mechanistically, the proviral long terminal repeat (LTR) acts as a strong promoter/enhancer, driving aberrant STAT3 transcription. The upregulated STAT3 pathway promotes cell survival and proliferation by increasing expression of anti-apoptotic genes (e.g., *BCL2* family members) and cytotoxic effectors, conferring a selective growth advantage to the affected T cell clone. This clonal expansion is a key feature of HIV-1 persistence and lymphomagenesis [41].

### 5.4. Effects of STAT3 Inhibition in HIV-1 Infected PBMCs

Ibba et al. investigated the role of STAT3 in HIV-1 infection by exposing PBMCs isolated from 20 healthy donors to HIV-1 in the presence or absence of a specific STAT3 inhibitor (Stattic^®^, also known as STAT3 inhibitor V) [66]. They observed that treatment with the STAT3 inhibitor consistently reduced HIV-1 replication, as measured by p24 levels in culture supernatants.

Gene expression analysis revealed that STAT3 inhibition significantly upregulated several immune-related genes known to exert anti-HIV-1 effects, including *CCL3*, *IL-1β*, and *IFN-α*. Notably, IL-12B mRNA expression increased by 23-fold, whereas IL-6 expression was markedly decreased. Although IL-6 has been previously reported to exert protective effects against HIV-1 infection [89,90], its downregulation in this context was unexpectedly associated with reduced viral replication. This paradox suggests that the antiviral response may rely more heavily on alternative immune mechanisms when STAT3 is inhibited.

These findings support the potential of STAT3 inhibitors as novel therapeutic agents against HIV-1 infection. However, caution is warranted, as STAT3 also plays roles in immune homeostasis and mucosal protection. Therapeutic inhibition of STAT3 must therefore balance antiviral effects with potential risks of exacerbating immune dysfunction.

### 5.5. Conclusive Considerations on the Role of STAT3 in HIV-1 Infection

In summary, HIV-1 modulates STAT3 through a combination of direct viral protein interactions, cytokine-mediated signaling, insertional mutagenesis, and manipulation of post-translational modifications. These mechanisms collectively contribute to immune dysregulation, viral persistence, and increased risk of malignancy in HIV-1-infected individuals. These findings highlight the multifaceted and cell–type–specific role of STAT3 signaling in the context of HIV-1 infection. In dendritic cells, HIV-1 gp120 enhances STAT3 activation via the IL-6 axis, potentially promoting an environment favorable to viral persistence. Conversely, in Th17 cells, HIV-1 impairs IL-23-induced STAT3 phosphorylation, leading to impaired cell survival and function. Interestingly, pharmacological inhibition of STAT3 in PBMCs resulted in reduced HIV-1 replication and enhanced expression of several antiviral genes, despite downregulation of IL-6. These apparently contrasting effects underscore the context-dependent nature of STAT3 activity and suggest that its role in HIV-1 pathogenesis varies across different immune cell subsets. Therefore, therapeutic strategies targeting STAT3 must carefully consider these divergent effects to avoid unintended immune suppression or enhancement of viral persistence.

## 6. Indirect Evidence for STAT4 in HIV-1 Pathogenesis

STAT4 is a key transcription factor orchestrating antiviral immunity [91,92]. It is primarily activated by IL-12, which drives the differentiation of CD4+T cells into Th1 cells, central mediators of cell-mediated immunity and major producers of IFN-γ [14,30,91]. In addition to this canonical nuclear function, STAT4 exerts non-canonical cytoplasmic activity by enhancing retinoic acid-inducible gene I (RIG-I)–mediated production of type I IFNs. RIG-I signaling is a pivotal component of the innate immune defense against RNA viruses. Once activated by the dsRNA, the N-terminus caspase activation and recruitment domains (CARDs) migrate and bind with CARDs attached to mitochondrial antiviral signaling protein (MAVS) to activate the signaling pathway for type I IFNs [92]. Silencing of STAT4 impairs IFN-β production in macrophages upon RNA virus infection, whereas overexpression of STAT4 enhances RIG-I-induced IFN-β promoter activation and IFN-stimulated response element activity [92].

While research has largely focused on STAT1, STAT3, and STAT5 [93,94,95], indirect evidence supports a potential involvement of STAT4 in anti-HIV-1 responses. Given its central role in Th1 polarization, any disruption of STAT4 signaling may compromise effective antiviral immunity and promote persistent immune activation in people living with HIV-1 [14].

A functional impairment of STAT4 can alter the Th1/Th2 balance, favoring a shift toward a Th2 profile. This skewing—frequently observed in HIV-1 infection [35]—reduces IFN-γ–mediated antiviral activity and contributes to immune exhaustion. Such dysregulation may result from direct viral effects or from changes in the cytokine milieu, such as diminished IL-12 production by dendritic cells, thereby limiting Th1 development [14,30]. IL-12–driven STAT4 activation is also required for optimal signaling through multiple cytokine pathways that converge on IFN-γ induction [96].

IFN-γ plays a multifaceted and context-dependent role in HIV-1 pathogenesis. Initially produced to facilitate the clearance of primary infection, IFN-γ contributes to the establishment of chronic immune activation. Unlike type I interferons, IFN-γ exhibits minimal direct antiviral activity against HIV-1 in primary cell cultures. Nevertheless, it enhances cytotoxic T lymphocyte and natural killer cell functions against HIV-1–infected cells, thereby supporting the control of viral replication. Conversely, in certain settings—particularly within mononuclear phagocytes—IFN-γ may paradoxically promote viral replication. This pro-viral effect is thought to result from its ability to sustain immune activation and induce inflammatory cytokines, thereby generating a more permissive environment for HIV-1 replication [97].

At the molecular level, HIV-1 interferes with the regulation of SOCS proteins, particularly SOCS1 and SOCS3, which normally restrain JAK/STAT activity and prevent excessive immune activation [98]. By downregulating SOCS expression, HIV-1 sustains STAT activation, fueling chronic immune stimulation and inflammation, hallmarks of disease progression. Although these effects have been studied mainly in relation to STAT1 and STAT3, evidence suggests that SOCS3 can directly inhibit STAT4 activation [99].

Genetic polymorphisms in STAT4 add another layer of complexity. The rs7574865 variant, for example, is associated with increased STAT4 expression and IFN-γ production and has been linked to autoimmune diseases such as systemic lupus erythematosus, rheumatoid arthritis, and autoimmune thyroiditis [100,101,102,103]. These variants may influence STAT4 transcription, phosphorylation, Th1/Th2 polarization, and cytokine responsiveness, with possible implications for immune control of HIV-1. Although most studies on STAT4 polymorphisms focus on autoimmunity, their potential role in HIV-1 infection warrants investigation, particularly in light of STAT4 involvement in Th1 differentiation and its regulation by SOCS3.

Beyond Th1 responses, STAT4 also influences CD4+ effector T-cell migration and contributes to the pathogenic potential of Th17 cells [104]. Both processes may exacerbate chronic inflammation and immune dysregulation in HIV-1 infection, underscoring the multifaceted nature of STAT4’s potential involvement in HIV pathogenesis.

## 7. Dysregulation of STAT5 Signaling in HIV-1 Infection: Implications for Immune Dysfunction and Viral Persistence

STAT5 is a transcription factor composed of two paralogs, STAT5A and STAT5B, which are primarily activated through the JAK/STAT signaling pathway in response to interleukins (e.g., IL-2, IL-3, IL-5, IL-7, IL-15, IL-21), erythropoietin, thrombopoietin, growth hormone, and prolactin [1,15,19,20,21,22,23]. The critical tyrosine residue for STAT5 activation is Tyr694 in STAT5A and Tyr699 in STAT5B. This phosphorylation is essential for dimerization, nuclear translocation, and DNA-binding activity [15]. STAT5 regulates the transcription of genes involved in proliferation, differentiation, and immune functions [15,22,29,32]. Mutations that cause dysregulated STAT5 signaling, or hyperactivation of the JAK/STAT pathway, contribute to tumor progression and immune dysfunction [15,19,20,21]. From a therapeutic perspective, STAT5 inhibition represents a promising target for the development of novel anticancer and immunomodulatory agents [15,16,17,18]. During HIV-1 infection, STAT5 signaling becomes markedly dysregulated, contributing both to immune dysfunction and to the persistence of the viral reservoir (Table 2).

### 7.1. Dual Roles of STAT5 Dysregulation in HIV-1 Infection: From Altered Cytokine Responsiveness to Viral Replication, and Clonal Expansion

A useful framework for understanding the complex impact of HIV-1 on STAT5 signaling is to distinguish between studies assessing the activation state of the pathway and those evaluating pathway responsiveness. Activation state studies consistently report increased levels of phosphorylated STAT5 following in vitro exposure of CD4+ T-cell and monocytic cell lines to HIV-1 [57], as well as constitutive activation of a truncated isoform of STAT5 (STAT5Δ) in the majority of HIV-1–infected individuals, particularly within CD4+ T cells [94]. Conversely, several studies have reported an impaired responsiveness of the STAT5 pathway to cytokine stimulation. In HIV-1-infected adults, CD8+ T cells display reduced phosphorylation of STAT5 isoforms in response to IL-2 [105], and a similar defect has been observed in HIV-1–infected monocyte-derived macrophages (MDMs) following GM-CSF stimulation [106].

Despite extensive evidence of cell signaling alterations induced by HIV-1 in vitro, the relevance of these changes to the clinical and/or immunologic status of HIV-1-infected individuals is often unclear. Lee et al. determined the JAK/STAT signaling changes at the single-cell level within distinct cell subsets from the primary immune cells of HIV-1-infected donors. They identified a specific defect in GM-CSF-driven STAT5 phosphorylation in the monocytes of HIV-1 donors despite normal GM-CSF receptor levels. In addition, they observed an enhancement of MAPK signaling associated with HIV-1 infection. This altered monocyte signaling response contributes to defective antigen presentation during HIV-1 infection [95].

Beyond immune signaling, STAT5 is also implicated in HIV-1 replication dynamics. Upon activation by γc cytokines such as IL-2, IL-7, and IL-15, STAT5 translocates to the nucleus, where it binds consensus sequences within the HIV-1 LTR, enhancing viral transcription. In primary CD4+ T cells, STAT5 overexpression increases both the frequency of HIV-1 p24+ cells and overall viral protein production, indicating that STAT5 activation promotes a permissive environment for viral replication [107].

Interestingly, the constitutively active truncated isoform STAT5Δ—prevalent in individuals with progressive HIV-1 infection—functions in an opposing manner. The STAT5Δ isoform arises through specific proteolytic activity that removes the C-terminal portion of STAT5, resulting in a constitutively active protein, particularly in myeloid cells and monocytes of HIV-positive patients. The main consequence of STAT5Δ expression in HIV infection is the negative regulation of HIV-1 transcription [94]. When complexed with NF-κB1/p50, STAT5Δ binds the HIV-1 LTR and inhibits RNA polymerase II recruitment, thereby contributing to viral latency [108,109]. The sustained presence of STAT5Δ during chronic infection may therefore reflect an immune adaptation that promotes latency at the expense of immune functionality.

Adding further complexity to this regulatory landscape, HIV-1 infection disrupts IL-7 signaling by impairing the nuclear translocation of phosphorylated STAT5. Landier et al. investigated IL-7 signaling in CD4+ T cells from viremic patients and healthy blood donors, using intracellular flow cytometry to monitor phosphorylation of STAT5 at the regulatory serine residue S726 and the critical tyrosine residue Y694. Their analysis revealed that progressive HIV infection leads to hyperphosphorylation of both S726 and Y694 in naive CD4+ T cells. However, quantitative image analysis demonstrated a defect in the nuclear relocalization of both phospho-STAT5 forms, which correlated with elevated HLA-DR expression. These findings suggest that chronic immune activation contributes to the disruption of IL-7 signal transduction [110].

STAT5 dysregulation in HIV-1-infected cells is also implicated in the clonal expansion of infected cells. In hematopoietic cells, HIV-1 preferentially integrates within the transcription unit of expressed genes and may induce aberrant RNA splicing mechanisms leading to the formation of chimeric transcripts harboring HIV sequences fused to cellular exon sequences. Moreover, HIV-1 and lentiviruses with active LTRs can effectively integrate and activate cancer-related genes, such as through promoter insertion, and thus induce uncontrolled clonal expansion. Cesana et al. identified HIV-1-mediated insertional activation of *STAT5B* and *BACH2* in Treg cells, resulting in the formation of chimeric mRNAs that drive overexpression of these genes and promote clonal expansion of infected cells. This mechanism, observed in nearly one-third of individuals on long-term ART, was particularly enriched in Treg and central memory T cells [111].

Follow-up studies using CRISPR/Cas9 modeling further demonstrated that HIV-1 LTR-driven expression of *BACH2* enhances proliferation and survival of Treg-like cells without compromising their suppressive function [112]. Additional work has shown that HIV-1 integration at the *BACH2* locus is recurrent in resting CD4+ T cells during suppressive ART, and that HIV-1-driven aberrant splicing may enhance host gene expression at the site of integration, contributing to clonal persistence [113,114].

### 7.2. The STAT5–CCR5 Axis: Mechanistic Insights and Implications for HIV-1 Cure Approaches

Bone marrow transplantation (BMT) has demonstrated the potential to cure HIV-1 infection, though such cases remain exceptionally rare. The most prominent examples include Timothy Ray Brown, the “Berlin Patient,” and the “London Patient,” both of whom received allogeneic hematopoietic stem cell transplants from donors carrying the homozygous CCR5-Δ32 mutation, a naturally occurring 32-base pair deletion in the CCR5 gene that renders the CCR5 receptor nonfunctional and confers resistance to HIV-1 infection [115,116]. These landmark cases underscore the therapeutic relevance of CCR5 and suggest that strategies aimed at replicating the effects of the CCR5-Δ32 mutation—such as gene editing—may hold promise for achieving HIV-1 remission or cure.

CCR5 is a G protein-coupled receptor predominantly expressed on immune cells, including CD4+ T cells, macrophages, dendritic cells, and microglia. It plays a critical role in immune surveillance by mediating leukocyte chemotaxis toward sites of inflammation in response to ligands such as CCL3 (MIP-1α), CCL4 (MIP-1β), and CCL5 (RANTES). Beyond its function in HIV-1 entry, CCR5 is implicated in a wide range of pathological conditions, including cancer and chronic inflammatory diseases such as rheumatoid arthritis, multiple sclerosis, and atherosclerosis, further highlighting its value as a therapeutic target [117,118,119,120].

In HIV-1 infection, CCR5 serves as a major co-receptor facilitating viral entry into CD4+ T cells. After initial binding to the CD4 molecule, HIV-1 engages CCR5 to promote membrane fusion and viral entry. Individuals homozygous for the CCR5-Δ32 allele lack functional surface expression of CCR5 and are highly resistant to HIV-1 acquisition [121].

The expression of CCR5 is tightly regulated at the transcriptional level. The transcription factor CREB-1 (cAMP response element-binding protein 1) activates *CCR5* gene expression by binding to the cAMP response element (CRE) in its promoter region following cAMP signaling [122]. Additional regulation involves the transcription factors Oct-1 and Oct-2, which recognize the same octamer motif but exert opposite effects: Oct-1 acts as a repressor, while Oct-2 enhances CCR5 expression [123]. Other factors such as NF-AT and GATA-1 have also been implicated in modulating *CCR5* transcription, although their roles may vary across cell types and activation states.

Recent evidence has revealed that the JAK/STAT signaling pathway plays a central role in regulating CCR5 expression. JAK inhibitors such as tofacitinib and ruxolitinib have been shown to reduce CCR5 surface levels on CD4+ T cells in vitro, leading to decreased HIV-1 production and reduced levels of integrated proviral DNA. This effect is likely mediated through inhibition of STAT5 phosphorylation. Importantly, elevated STAT5 activity is associated with higher levels of HIV-1 integration in infected individuals, suggesting that its inhibition could alter the cellular environment necessary for HIV-1 persistence [50].

The role of STAT signaling is particularly relevant in the context of interleukin-15 (IL-15), a γ-chain cytokine that promotes survival and proliferation of memory CD4+ T cells and NK cells, two key components of the latent HIV-1 reservoir and antiviral immune response, respectively [35]. IL-15 can induce reactivation of latent HIV-1 in memory CD4+ T cells, leading to viral gene expression and the release of infectious virions. This reactivation not only increases viremia but also facilitates infection of bystander activated CD4+ T cells, perpetuating the viral reservoir [124].

IL-15 exerts its effects through a receptor complex composed of IL-15Rα, IL-2/IL-15Rβ (CD122), and the common γ-chain (CD132), which together activate the JAK/STAT pathway—particularly STAT5—promoting both immune activation and HIV-1 reactivation. JAK inhibitors suppress this IL-15-mediated STAT activation, thereby blocking reactivation of latent HIV-1 and reducing the potential for reservoir replenishment [125].

Further insights into the role of JAK/STAT signaling in regulating CCR5 have come from the work of Wang et al., who demonstrated that JAK/STAT inhibitors—including tofacitinib, ruxolitinib, baricitinib, IQDMA, FLLL32, and WP1066—consistently reduced CCR5 and CCR2 expression at both the mRNA and protein levels in primary human CD4+ T cells [126]. Knockout studies revealed that loss of individual genes such as JAK2, STAT3, STAT5A, or STAT5B significantly downregulated CCR5 and CCR2 expression, while combined knockouts (e.g., STAT3/STAT5 or JAK2/STAT) produced an even greater reduction [126]. Although CCR2 is not a co-receptor for HIV-1, it plays an important role in immune cell recruitment via its ligand CCL2 (MCP-1), particularly to lymphoid tissues where active HIV-1 replication and reservoir formation occur [127,128]. Moreover, monocytes and macrophages that are CCR2+ cells serve as important HIV-1 reservoirs and contribute to viral persistence [129].

Taken together, these findings strongly support a central role for the JAK/STAT signaling pathway—especially STAT5—in modulating CCR5 expression and shaping the cellular environment conducive to HIV-1 persistence. This regulatory axis provides a compelling therapeutic target for limiting viral entry, reactivation from latency, and long-term reservoir maintenance.

### 7.3. STAT5 at the Crossroads of HIV-1 Latency and Persistence

Despite the effectiveness of ART in suppressing viral replication, a small subset of cells harboring integrated, replication-competent HIV-1 DNA remains, constituting a major barrier to cure [35]. This persistent infection is largely maintained within central memory (TCM), transitional memory (TTM), and effector memory (TEM) CD4+ T cells, whose long-term survival depends on cytokines engaging the common γ-chain receptor [35,50]. Among these cytokines, IL-2, IL-7, and IL-15 are particularly important, as they sustain T cell memory and activate STAT5-mediated signaling [50,130,131].

STAT5 exerts a context-dependent influence on HIV-1 biology. On one hand, STAT5 activation generally promotes viral replication, as discussed in Section 7.1. On the other hand, STAT5-driven proliferation and survival indirectly contribute to the persistence and expansion of the latent reservoir, particularly in regulatory and memory T cells, where STAT5B signaling supports the long-term maintenance of infected cells (Figure 5). In addition, the STAT5Δ isoform directly favors HIV-1 latency by acting as a negative regulator of viral transcription, especially in myeloid cells and monocytes. Preliminary evidence suggests that dendritic cells may also express STAT5Δ, although its role in these cells remains less well defined. By contrast, STAT5Δ expression is considerably lower in CD4+ T lymphocytes, and in vivo studies indicate that it is scarcely detectable in lymphocytes compared with monocytes [94,108,109]. These observations support the view that STAT5Δ contributes mainly to latency regulation in the myeloid compartment, thereby adding a distinct layer of complexity to the interplay between HIV-1 and host transcriptional control.

The dual role of STAT5—facilitating viral replication while promoting reservoir expansion—is exemplified by the study of De Armas et al., who demonstrated that the JAK1/2 inhibitors ruxolitinib and baricitinib significantly curtailed both latent and productive infection when administered 24 h post-infection in tonsillar CD4+ T cells. Their findings further underscored the STAT pathway as a key regulator of reservoir stability and reactivation potential. Notably, they reported a dose-dependent reduction in T cell activation markers: CD25 expression declined by more than 80% at the highest drug concentrations, while HLA-DR and CD69 were reduced by approximately 60% and 50%, respectively [130].

In line with these findings, Janssens et al. showed that small-molecule JAK/STAT inhibitors suppress HIV-1 reactivation ex vivo. Screening of 26 compounds in PBMCs from ART-suppressed individuals revealed that JAK/STAT inhibitors interfere with transcriptional initiation and elongation, thereby reducing viral RNA production upon T cell activation. Several agents demonstrated broad inhibitory effects across multiple transcriptional checkpoints, reinforcing the view that STAT signaling is integral to both maintenance and reversal of latency [132].

Beyond latency maintenance, the STAT system also mediates the effects of latency-reversing agents (LRAs). Systems biology approaches have identified JAK/STAT signaling as a key node in the transcriptional network driving HIV-1 reactivation, acting in concert with NF-κB, JNK, and ERK pathways. STAT activation is particularly important in the immediate early phase of latency reversal, positioning it as a potential dual-use target—capable of limiting reservoir persistence while modulating the efficacy of “shock and kill” strategies [131].

In summary, the role of STAT5 in HIV-1 latency is inherently dual and, at times, contradictory. While STAT5 activation can promote viral transcription in response to immune stimuli—a property exploitable in latency reversal—its survival and proliferative signals also support the persistence of memory CD4+ T cells, the main reservoir of latent HIV-1. This paradox underscores the need for therapeutic balance: strategies that harness STAT5 for latency reversal must avoid inadvertently promoting reservoir stability.

## 8. STAT6 as a Host Factor Manipulated by KSHV: Implications for Kaposi’s Sarcoma in HIV-1 Infection

STAT6 is a transcription factor primarily activated by IL-4 and IL-13, playing a central role in Th2 cell differentiation, B cell activation, alternative (M2) macrophage polarization, and humoral immunity [14,133,134].

Beyond classical cytokine signaling, STAT6 is also activated in response to viral nucleic acids via the STING-TBK1 pathway, independent of JAKs. This non-canonical activation leads to STAT6 phosphorylation, dimerization, and nuclear translocation, inducing genes involved in immune cell homing and antiviral defense. However, viruses can subvert this pathway by interfering with STAT6 post-translational modifications, nuclear translocation, or promoting its degradation, thereby evading innate immune responses, although this has not been specifically delineated in HIV-1 pathogenesis in vivo [135,136,137]. To date, no studies have specifically addressed whether HIV-1 infection directly alters STAT6 activation, signaling, or regulation, and thus evidence on this point is currently lacking in the scientific literature. Nevertheless, given the clinical relevance of Kaposi’s sarcoma (KS) as a major complication of HIV-1 infection, and considering that STAT6 has been implicated in the regulation of Kaposi sarcoma–associated herpesvirus (KSHV) activity, it is important to discuss STAT6 in this context. For this reason, the following section will focus on STAT6 in relation to KSHV-driven oncogenesis in HIV-1–infected individuals, as this provides a mechanistic and clinically relevant link to the broader theme of HIV-1 pathogenesis. HIV-1 infection markedly increases the risk of KS in persons infected with KSHV. KSHV is the necessary cause of KS, and HIV-related immunosuppression and inflammation act as key cofactors that amplify KS risk and aggressiveness. Intracellular HIV-1 Tat protein can activate the lytic cycle replication of KSHV in BCBL-1 cells, primarily through modulation of the JAK/STAT signaling pathway [138]. Tat expression led to increased production of IL-4, which in turn activates STAT6. Functional experiments using neutralizing antibodies and pathway inhibitors revealed that blockade of IL-4 or inhibition of STAT6 signaling partially reduced Tat-induced KSHV lytic gene expression, indicating that STAT6 activation plays a role in supporting KSHV reactivation. However, Wang et al. observed that KSHV manipulates STAT6 to promote viral latency. The virus-encoded latency-associated nuclear antigen (LANA) interacts directly with STAT6, driving its nuclear localization independently of canonical tyrosine 641 phosphorylation. Within the nucleus, LANA induces a serine protease-dependent cleavage of STAT6, generating a truncated form lacking the transactivation domain. This cleaved STAT6 acts as a dominant-negative regulator, repressing transcription of the KSHV replication and transcription activator (RTA), which is essential for lytic reactivation. As a result, the presence of nuclear cleaved STAT6 enforces maintenance of viral latency by suppressing the lytic cycle. Functional studies show that knockdown of STAT6 leads to increased RTA expression, enhanced lytic reactivation [139].

In summary, Tat-mediated lytic activation increases the risk and severity of Kaposi’s sarcoma, whereas LANA/STAT6–mediated latency promotes the survival of infected cells and viral persistence. The interplay between these two mechanisms drives clinical progression: predominance of the lytic phase leads to rapid Kaposi’s sarcoma development and greater immune compromise, while latency favors viral persistence and therapeutic resistance, contributing to disease recurrence and chronicity. This highlights STAT6 as a key host factor subverted by KSHV to regulate the balance between latency and lytic replication, with implications for viral persistence and pathogenesis of KS in HIV infected individuals.

## 9. Conclusions

The STAT family of transcription factors plays a central role in orchestrating host immune responses, particularly during viral infections. Among the seven mammalian STAT proteins, STAT1, STAT3, and STAT5 are most prominently affected by HIV-1, and their dysregulation drives immune dysfunction, viral persistence, and latency.

HIV-1–mediated alterations in STAT signaling contribute to chronic, yet ineffective, immune activation. Persistent STAT3 activation in dendritic cells, combined with STAT5 downregulation in CD8+ T cells and macrophages, further impairs immune function, while STAT5 activation in CD4+ T cells enhances viral replication. Notably, HIV-1 inhibits IL-23–driven STAT3 activation in Th17 cells, reducing IL-17 production, compromising intestinal mucosal barrier integrity, and promoting microbial translocation that fuels chronic immune activation. These changes also disturb the Th17/Treg balance and increase susceptibility to opportunistic infections. STAT5 additionally supports viral latency, particularly through the truncated STAT5Δ isoform.

Together, these findings highlight how HIV-1 exerts divergent, context-dependent effects on the STAT pathway to evade immunity and maintain persistent infection. This complexity presents both challenges and opportunities for therapeutic intervention. While JAK/STAT inhibitors, such as ruxolitinib, show potential as adjunctive treatments, clinical evidence of their efficacy in improving HIV-1 outcomes remains limited, and their broad immunomodulatory effects may carry unintended risks [140,141]. Moreover, these findings suggest that the immunological assessment of HIV-1–infected patients should not be limited to CD4+ T-cell counts and the CD4/CD8 ratio alone. A more comprehensive approach should also include the functional analysis of intracellular signaling pathways most affected by the virus. In particular, the evaluation of basal and phosphorylated STAT1, STAT3, and STAT5 in CD4+ and CD8+ T-cell subsets and in monocytes—using flow cytometry, for example—may provide additional insights into the degree of immune dysfunction and the extent of residual chronic inflammation. Integrating these parameters into clinical practice could not only refine the immunological stratification of patients but also pave the way for personalized therapeutic strategies that specifically address the HIV-1–induced alterations of the JAK/STAT network.

In conclusion, HIV-1 manipulation of the STAT signaling network exemplifies a sophisticated viral strategy to subvert host immunity. Therapeutic targeting of this pathway offers a promising avenue, but a deeper understanding of isoform-specific roles and cell-type–specific dynamics will be essential to safely and effectively harness JAK/STAT modulation in HIV-1 infection and cure strategies.

## Figures and Tables

**Figure 1 ijms-26-09123-f001:**
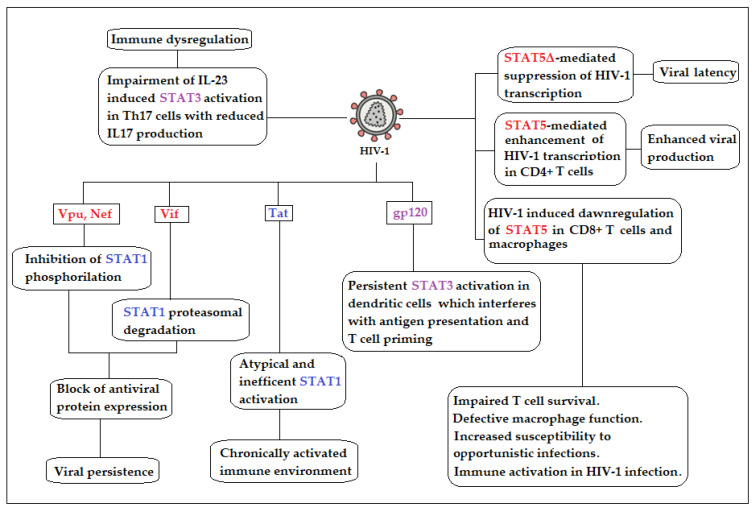
Schematic representation of the multifaceted role of the STAT pathway in HIV-1 infection. HIV-1 accessory proteins Vpu and Nef inhibit STAT1 activation or promote its proteasomal degradation, thereby preventing STAT1-mediated induction of genes encoding key antiviral proteins. In addition, the viral protein Tat triggers an atypical and inefficient activation of STAT1, which sustains chronic immune activation without effectively contributing to HIV-1 clearance. The state of immunodeficiency is further exacerbated by persistent STAT3 activation in dendritic cells, which interferes with antigen presentation and T-cell priming, together with an impaired IL-23–induced STAT3 activation in Th17 cells, leading to reduced IL-17 production. The modulation of STAT5 by HIV-1 is even more complex. On one hand, infected cells produce a truncated STAT5 isoform (STAT5Δ) that suppresses HIV-1 transcription and promotes viral latency. On the other hand, STAT5 enhances proviral transcription in CD4+ T cells, thereby increasing virion production and viral spread. Conversely, in CD8+ T cells and macrophages, HIV-1 induces a downregulation of STAT5, which further compromises the immune system and promotes a state of chronic but ineffective immune activation that fails to eliminate the virus. HIV-1 appears to exert only indirect and potentially uncertain effects on STAT4 and STAT6, which is why they have been omitted from this schematic representation. STAT2 can dimerize with STAT1 to form the ISGF3 complex. No specific or relevant data regarding STAT2 and HIV-1 are reported in the literature.

**Figure 2 ijms-26-09123-f002:**
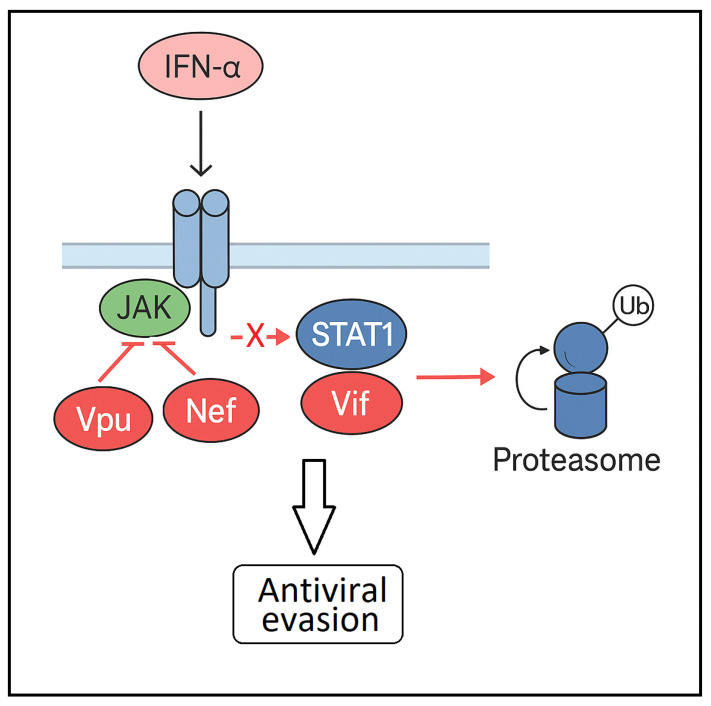
HIV-1 protein Vpu and Nef interfere with the kinase activity of JAK-STAT1, inhibiting the phosphorylation/activation of STAT1. In addition, Vif interacts with STAT1, promoting its ubiquitination and subsequent proteasomal degradation. These effects avoid antiviral protection induced by IFN-α.

**Figure 3 ijms-26-09123-f003:**
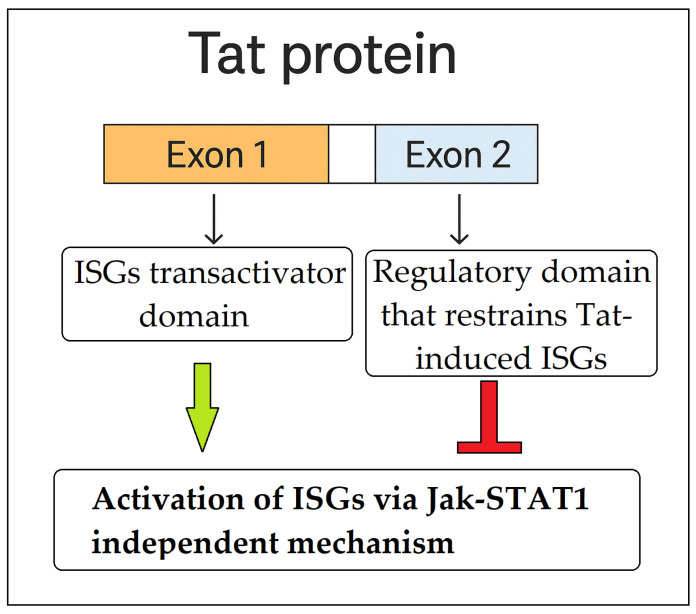
HIV-1 Tat directly regulates the expression of ISGs in antigen-presenting cells by interacting with specific gene promoters and transcription factors, bypassing the canonical IFN-induced JAK-STAT pathway. Tat exon 2 functions as a regulatory domain that restrains Tat-induced ISGs activation. This implies that HIV-1 fine-tunes ISG expression through Tat to avoid initiating a robust antiviral state typically triggered by IFNs.

**Figure 4 ijms-26-09123-f004:**
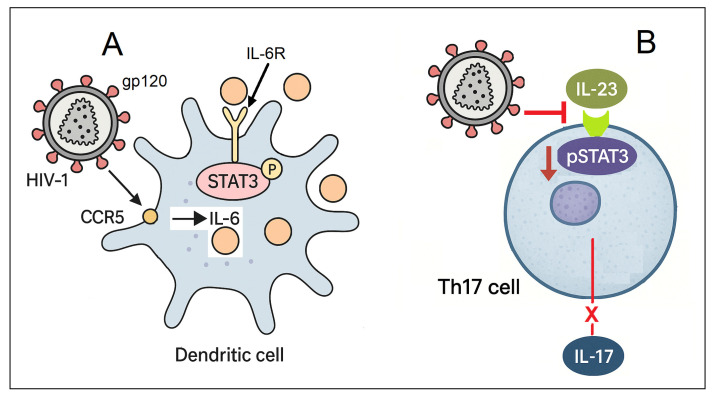
(**A**) HIV-1 gp120 interacts with the CCR5 receptor on dendritic cells, triggering the production of interleukin-6 (IL-6). The secreted IL-6 then engages its receptor in an autocrine manner, activating STAT3 and establishing a classical positive feedback loop. Persistent STAT3 activation in dendritic cells interferes with antigen presentation and T cell priming. (**B**) Concurrently, HIV-1 impairs IL-23-induced STAT3 phosphorylation in Th17 cells, leading to significantly reduced STAT3 activation and diminished IL-17 production.

**Figure 5 ijms-26-09123-f005:**
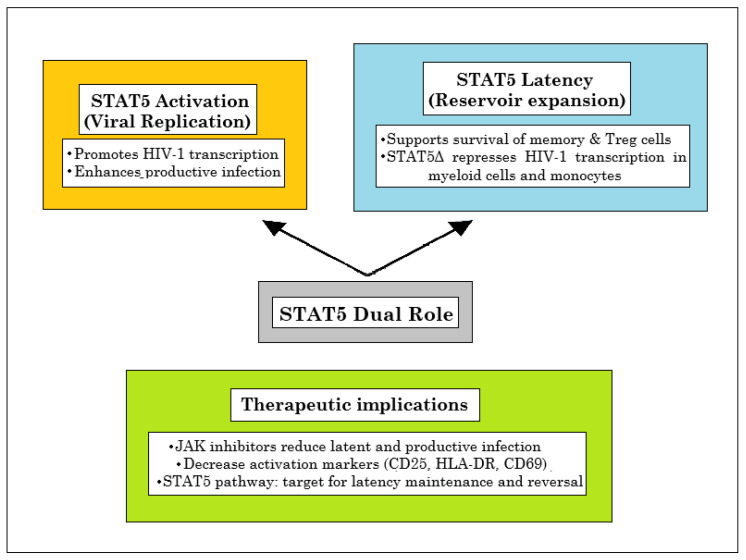
The dual role of STAT5 in viral replication and reservoir expansion.

**Table 1 ijms-26-09123-t001:** STAT1-induced antiviral proteins and mechanisms against HIV-1.

Protein	Antiviral Mechanism Against HIV-1	References
APOBEC3G	Cytidine deamination to uracil (hypermutation); inhibition of reverse transcriptase.	[53]
Tetherin (BST2)	Blocks release of HIV-1 virions from cell surface	[53]
SAMHD1	Depletes dNTPs, restricting reverse transcription	[53]
MX2 (MXB)	Inhibits nuclear import of HIV-1 pre-integration complex	[53]
GBP5	Inhibits HIV-1 envelope processing and infectivity	[53]
Schlafen 11 (SLFN11)	Inhibits HIV-1 protein synthesis by tRNA restriction	[53]
TRIM56	Enhances ISG induction, inhibits late HIV-1 gene expression	[54]
IDO1	Depletes tryptophan, suppressing HIV-1 replication	[54]
IRF-1	Transcription factor, suppresses HIV-1 LTR-driven gene expression	[54]
ISG15	Ubiquitin-like modifier, modulates immune signaling and restricts HIV-1	[12]

**Table 2 ijms-26-09123-t002:** Modulation of the STAT5 signaling pathway by HIV-1 infection.

Aspect	Description	Cell Types Involved	References
STAT5 in HIV-1 infected cells	Increased STAT5 phosphorylation following HIV-1 exposure in vitro.	CD4^+^ T cells, monocytes.	[57,94]
Altered monocyte/macrophage function	Impaired GM-CSF-induced STAT5 phosphorylation and enhanced MAPK signaling contribute to defective antigen presentation.	Monocytes and macrophages.	[95]
Impaired cytokine responsiveness	Reduced STAT5 phosphorylation in response to IL-2 (CD8+ T cells) and GM-CSF (macrophages).	CD8^+^ T cells, macrophages.	[105,106]
Role in viral replication	Full-length STAT5, activated by IL-2, IL-7, or IL-15, enhances HIV-1 LTR transcription and viral protein production (e.g., p24+ cells).	CD4^+^ T cells.	[107]
STAT5Δ (truncated isoform)	Constitutively active; binds the HIV-1 LTR and inhibits viral transcription by blocking RNA polymerase II recruitment.	Myeloid cells, monocytes.	[94,108,109]
Disrupted IL-7 signaling	Hyperphosphorylation of STAT5 at S726 and Y694, but defective nuclear translocation; correlates with elevated HLA-DR expression.	CD4^+^ T cells.	[110]
Insertional activation of STAT5B	HIV-1 integration in STAT5B and BACH2 driving clonal expansion.	Treg cells, central memory T cells.	[111,112,113,114]

## Data Availability

Not available.

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
