# Peer review of "The STAT Signaling Pathway in HIV-1 Infection: Roles and Dysregulation"

_ijms, 2025, doi:10.3390/ijms26189123_

Round 1
Reviewer 1 Report
Comments and Suggestions for Authors
This is an easy review to read and will direct readers interested in STAT activity in HIV-1 infection to relevant primary research articles. As the authors point out in their article, reports of STAT regulation in the context of HIV infection can be confusing and often contradictory to other reports. However, STAT activity is critical for survival, differentiation, and antiviral pathways; making this article's topic highly relevant to the field. My only suggestion is that there is a typo in Figure 1: "dawn-regulation"
Author Response
Comments 1: This is an easy review to read and will direct readers interested in STAT activity in HIV-1 infection to relevant primary research articles. As the authors point out in their article, reports of STAT regulation in the context of HIV infection can be confusing and often contradictory to other reports. However, STAT activity is critical for survival, differentiation, and antiviral pathways; making this article's topic highly relevant to the field. My only suggestion is that there is a typo in Figure 1: "dawn-regulation".
Response 1: We thank the reviewer for noticing this typo. The term “dawn-regulation” in Figure 1 has now been corrected.
Reviewer 2 Report
Comments and Suggestions for Authors
This manuscript, entitled "The STAT Signaling Pathway in HIV-1 Infection: Roles and Dysregulation", summarizes the evolution, latest advances, and future perspectives of STAT signaling pathway in HIV-1 infection. The manuscript is comprehensive and timely, covering recent study and providing useful insights for readers. Nevertheless, several issues require major revision before the paper can be published.
1.The authors have summarized the roles of each STAT protein, but notably omitted STAT2, which should be supplemented. If there is no relevant research available, it should also be explicitly stated in the article.
2.Regarding STAT6, the authors' summary focuses more on KSHV rather than HIV, which deviates from the main theme of this article. It is suggested that the section on STAT6 should be more focused on the research progress related to HIV.
Author Response
Comments 1: The authors have summarized the roles of each STAT protein, but notably omitted STAT2, which should be supplemented. If there is no relevant research available, it should also be explicitly stated in the article.
Response 1: We thank the reviewer for this comment. STAT2 is indeed included in Figure 1, where we state: “No specific or relevant data regarding STAT2 and HIV-1 are reported in the literature.” In accordance with the reviewer’s suggestion, we have now also explicitly mentioned this in the main text (line 174) to clarify the lack of available data.
Comments 2: Regarding STAT6, the authors' summary focuses more on KSHV rather than HIV, which deviates from the main theme of this article. It is suggested that the section on STAT6 should be more focused on the research progress related to HIV.
Response 2: We thank the reviewer for this valuable observation. We have revised the section on STAT6 accordingly. In the updated version, we now explicitly state that, to the best of our knowledge, no studies have so far addressed possible alterations of STAT6 induced by HIV-1. However, given the clinical importance of Kaposi’s sarcoma (KS) as a major complication of HIV-1 infection and the involvement of STAT6 in the regulation of KSHV activity, we have clarified the rationale for including a discussion on STAT6 in this context. This modification ensures that the section remains aligned with the main theme of the manuscript while also addressing the link between HIV-1 infection, immune dysregulation, and KS development.
Reviewer 3 Report
Comments and Suggestions for Authors
This review is probably the best among those devoted to the role of STAT in HIV infection, even though it does not cite all reports on the topic, published to date. The authors give a detailed description of the contribution of STAT to HIV persistence and latency. In addition to analyzing mechanisms of STAT dysregulation associated with HIV infection, the review covers data obtained in cohorts of HIV/HCV-coinfected patients on the expression of type I interferons, which is one of the key strengths of this work.
The manuscript is very well illustrated by drawings and tables, which give an exhaustive summary of all relevant information and can be used as a self-sufficient study guide. Overall, the review will be useful in refining approaches to the treatment of HIV infection.
Author Response
Comments 1:
This review is probably the best among those devoted to the role of STAT in HIV infection, even though it does not cite all reports on the topic, published to date. The authors give a detailed description of the contribution of STAT to HIV persistence and latency. In addition to analyzing mechanisms of STAT dysregulation associated with HIV infection, the review covers data obtained in cohorts of HIV/HCV-coinfected patients on the expression of type I interferons, which is one of the key strengths of this work.
The manuscript is very well illustrated by drawings and tables, which give an exhaustive summary of all relevant information and can be used as a self-sufficient study guide. Overall, the review will be useful in refining approaches to the treatment of HIV infection.
Response 1: We would like to sincerely thank you for the time and attention you dedicated to reviewing our manuscript. Your comments are extremely encouraging, and in my career, no referee has ever given such a detailed and motivating compliment as yours: "This review is probably the best among those devoted to the role of STAT in HIV infection …”. Your evaluation not only honors us but also confirms that our work has achieved its goal: providing a comprehensive and useful analysis for the scientific community. Your positive remarks about the illustrations and tables, considered as self-sufficient study tools, are particularly encouraging and motivate us to continue in this direction.